# Immunization and Immunotherapy Approaches against *Pseudomonas aeruginosa* and *Burkholderia cepacia* Complex Infections

**DOI:** 10.3390/vaccines9060670

**Published:** 2021-06-18

**Authors:** Sílvia A. Sousa, António M. M. Seixas, Joana M. M. Marques, Jorge H. Leitão

**Affiliations:** 1Department of Bioengineering, IBB—Institute for Bioengineering and Biosciences, Instituto Superior Técnico, Universidade de Lisboa, Av. Rovisco Pais, 1049-001 Lisboa, Portugal; antonio.seixas@tecnico.ulisboa.pt (A.M.M.S.); joanammmarques@tecnico.ulisboa.pt (J.M.M.M.); 2Associate Laboratory, i4HB—Institute for Health and Bioeconomy at Instituto Superior Técnico, Universidade de Lisboa, Av. Rovisco Pais, 1049-001 Lisboa, Portugal

**Keywords:** *Pseudomonas aeruginosa*, *Burkholderia cepacia* complex, active immunization, passive immunotherapy, nosocomial infections, cystic fibrosis

## Abstract

Human infections caused by the opportunist pathogens *Burkholderia cepacia* complex and *Pseudomonas aeruginosa* are of particular concern due to their severity, their multiple antibiotic resistance, and the limited eradication efficiency of the current available treatments. New therapeutic options have been pursued, being vaccination strategies to prevent or limit these infections as a rational approach to tackle these infections. In this review, immunization and immunotherapy approaches currently available and under study against these bacterial pathogens is reviewed. Ongoing active and passive immunization clinical trials against *P*. *aeruginosa* infections is also reviewed. Novel identified bacterial targets and their possible exploitation for the development of immunization and immunotherapy strategies against *P*. *aeruginosa* and *B*. *cepacia* complex and infections are also presented and discussed.

## 1. *Pseudomonas aeruginosa* Infections and Their Current Management

*Pseudomonas aeruginosa* are opportunistic pathogens that can cause hospital-acquired infections, especially to critically ill, immunocompromised, and burn wound patients, as well as those with cystic fibrosis (CF) [1,2]. These bacteria can cause severe pulmonary infections that can lead to acute pneumonia (e.g., ventilator-associated pneumonia–VAP) or to chronic pneumonia (e.g., common in CF patients) [1].

The major problem associated with chronic infections and high mortality is the emergence of drug-resistant strains, due to intrinsic and acquired resistance mechanisms [3]. The emergence of multiple drug resistant (MDR), extensively drug-resistant (XDR) and pandrug-resistant (PDR) *P*. *aeruginosa* strains have been extensively reported [3]. In 2019, according to the EARS-Net report, rates of bacterial resistance in European countries, 31.8%, 17.6% or 12.1% of *P*. *aeruginosa* isolates were resistant to at least one, two or three antimicrobial groups under surveillance (i.e., piperacillin/tazobactam, fluoroquinolones, ceftazidime, aminoglycosides and carbapenems), respectively [4]. *P*. *aeruginosa* exhibits multiple resistance mechanisms to antibiotics including mutational resistome, horizontally acquired resistome, decreased membrane permeability, expression of efflux systems, production of antibiotic-inactivating enzymes and enzymes that perform target modification [3,5]. Recently, carbapenem-resistant *P*. *aeruginosa* were included by the World Health Organization in the “critical-priority” bacterial group for which new antibiotics are urgently needed [6]. Recent studies have also provided evidence of three MDR/XDR global clones (ST175, ST111 and ST235), referred to as “high-risk” clones, disseminated in several hospitals worldwide [5].

Early diagnosis of *P*. *aeruginosa* infections and immediate administration of an adequate antibiotic regimen has been associated with an improved clinical outcome [7]; conversely, late prescription of an adequate antibiotic therapy has been related to a significant increase in mortality. Currently, one of the most common approaches consists of an anti-pseudomonal β-lactam (piperacillin/tazobactam, ceftolozane/tazobactam, ceftazidime, cefepime or a carbapenem) combined with another anti-pseudomonal agent (aminoglycoside or a fluoroquinolone) [8]. Recently, Horcajada et al. (2019) reviewed the current treatments available and the efficacy results of new anti-pseudomonads compounds against MDR and XDR *P*. *aeruginosa* infections [5]. However, current therapies have limited results, therefore alternative drugs and new therapeutic options are still needed. One of the emerging strategies for combating these infections involves the use of vaccines or monoclonal antibodies for prevention of the acquisition of MDR *P*. *aeruginosa* infections by high-risk patients.

## 2. *Burkholderia cepacia* Complex Infections and Their Current Management

The *Burkholderia cepacia* complex (Bcc) is a group of Gram-negative and opportunistic bacteria that comprises at least 22 closely related but genetically distinct species [9]. The Bcc encompasses *Burkholderia cepacia* as the type species, as well as other species, including the two species responsible for most of the CF infections worldwide, *Burkholderia cenocepacia* and *Burkholderia multivorans* [10,11]. Bcc infections remain life threatening to patients with CF and chronic granulomatous disease (CGD), and more recently emerged as important pathogens among immunocompromised and hospitalized patients. Although a small percentage of CF patients are infected with Bcc worldwide (approx. 3.5%) [12], these infections are particularly feared due to their highly variable and unpredictable clinical outcomes, ranging from asymptomatic carriage to cepacia syndrome [13]. Patients infected with Bcc often develop chronic infection and persistent inflammation, leading to progressive lung damage and mortality [10,13]. Ultimately, many patients can develop cepacia syndrome, characterized by the rapid development of a fatal and necrotizing pneumonia with associated bacteremia [14]. Despite therapy, chronicity is developed in 94% (*n* = 33) and 50% (*n* = 13) of *B*. *cenocepacia* and *B*. *multivorans* infections, respectively [15]. Furthermore, infections caused by *B*. *cenocepacia* are usually associated with poorer clinical prognosis, higher transmissibility and mortality, while infections by *B*. *multivorans* have milder clinical manifestations and lower transmissibility and mortality [15]. Accordingly, in a murine model of pulmonary infection, a greater degree of illness was observed with *B*. *cenocepacia* strains when compared to *B*. *multivorans* [16].

The emergence of Bcc infections is known to be increased by patient-to-patient transmission, which is an important risk factor for colonization [17]. The transmission of Bcc strains among CF patients can occur both within and outside hospital settings and vary according to a number of factors including the strain, patient population and treatment center. Thus, the implemented strategies to effectively help prevent or delay infection usually include patient segregation and emphasis on hygiene of shared equipment [18]. General recommendations for infection prevention and control in CF patients infected with Bcc bacteria was recently reviewed by Saiman et al. (2014) [19].

Knowledge of the molecular mechanisms underlying pathogenicity and virulence of Bcc bacteria is critical for the development of new approaches for infection eradication in CF patients. The virulence and pathogenicity of Bcc is multifactorial, involving several virulence factors such as exopolysaccharide (cepacian), lipopolysaccharide (LPS), secretion systems, siderophores, extracellular proteases, flagellin and biofilm formation [13,20]. In addition, *B*. *cenocepacia* can survive intracellularly in airway epithelial cells and macrophages, to evade host defenses and cause chronic infections [21]. To adhere and invade the lung epithelial cells, these bacteria use adhesins, flagella, pili and lipases, whereas the type IV secretion system contributes to intracellular survival and replication of *B*. *cenocepacia* [21]. Unlike other CF pathogens, Bcc strains are able to cross the epithelial barrier, enter the blood stream and cause bacteremia.

Bcc are intrinsically resistant to several clinically available drugs, making the establishment of safe and effective therapies very difficult. Resistance can also be acquired due to a variety of mechanisms, such as reduced permeability of the cell envelope, increased efflux activity, mutations in the antibiotic target and enzymatic modification or inactivation of the antibiotic [22,23]. Moreover, these bacteria are resistant to disinfectants, antiseptics and pharmaceutical solutions, which also leads to outbreaks among CF and also non-CF hospitalized immunosuppressed patients [20,24].

The unpredictability of the Bcc infection outcome, together with the intrinsic and acquired resistance to antibiotics, the rapid patient-to-patient transmission and the capacity to adapt to environmental changes, makes the treatment of a chronic infection caused by Bcc difficult and challenging. As in the case of *P*. *aeruginosa*, to be effective, treatments require early, aggressive, prolonged and usually combined antibiotic therapy [23,25]. However, there is still no consensus regarding the duration of the treatment, the usage of single or combined antibiotic therapy and other aspects, in addition to the lack of correlation between in vitro and in vivo trials (reviewed by Gautam et al., 2015) [26].

## 3. Immunization Strategies against *P*. *aeruginosa* and Bcc Infections

Due to the importance of *P*. *aeruginosa* in nosocomial infections and in chronic infections of CF patients and their everlasting problem of antibiotic resistance, several population group candidates have been suggested for immunization with a *P*. *aeruginosa* preventive vaccine. These candidates include everyone over 60 years old, preoperative patients scheduled for major surgery, wearers of extended-use contact lenses and CF patients [27]. The immunization strategies against *P*. *aeruginosa* infections have been mainly focused on these important antigens: LPS, flagellar protein components and outer membrane protein F and I (OprF/I) [27], as depicted in Figure 1. Several other antigens have also been tested, however, only a few entered clinical trials (Table 1) [28].

Several LPS O-antigen based vaccines have been developed (Table 1) due to the observation that they can often elicit antibodies that are protective in animal models. However, protection is only obtained for the specific O-antigen serotypes of the strains used [29]. In the case of CF patients, it is known that *P*. *aeruginosa* isolates recovered in the early stages of disease are non-mucoid and LPS-smooth, while the late isolates, are mucoid and LPS-rough [30]. Mucoid strains produce the anionic exopolysaccharide alginate, which has been ascribed various functions related to the pathogenesis of *P*. *aeruginosa* in the CF lung. These include, among others, inhibition of phagocytosis by macrophages and neutrophils, suppression of neutrophil chemotaxis and opsonic antibody production, scavenging of hypochlorite and quenching of oxygen reactive species (reviewed in [31]). In addition, alginate contributes to biofilm formation, promoting the formation of microcolonies and contributing to antibiotic resistance [31].

In CF, the *P*. *aeruginosa* strains that colonize the patients are usually flagella-positive, which are classified as of the “a” or “b” serotypes [32]. Immunization with *P*. *aeruginosa* flagella has been shown to induce protection in various animal-infection models and could prevent either acute, chronic or in combination, infection in CF patients, although with only a small but statistically significant reduction in *P*. *aeruginosa* infection compared with the placebo-treated patients [33]. After vaccination, the tested *P*. *aeruginosa* positive patients had strains with other flagella types not included in the vaccine formulation, suggesting a positive effect of the vaccine. However, the IMMUNO company (Austria) responsible for the manufacture of this vaccine stopped production before the end of the trial.

Outer membrane proteins (OMP) are surface-exposed, highly conserved and immunogenic in *P*. *aeruginosa*, and are being considered as good vaccine candidates [34]. In *P*. *aeruginosa*, the most investigated OMPs are the major porin F (OprF) and the lipoprotein I (OprI) [35,36,37,38]. An OprF/I vaccine was shown to induce opsonic antibodies and antibodies that inhibit IFN-γ binding to *P*. *aeruginosa* [37]. The OprF/I vaccine shows promising results as an active vaccine and new formulations by combination with a Th17-stimulating antigen (e.g., exotoxin PopB) are being pursued.

The majority of the vaccines developed or under study act on a single target, thus lacking a broad range of protection or were not effective at all [39]. Recent studies have shown that vaccines with multiple antigens can significantly increase the immune response, thereby increasing their preventive potential [40,41]. Whole organisms, such as killed whole-cell and live attenuated *P*. *aeruginosa*, are also being tested for vaccine development [42,43]. Immunization with a multivalent live-attenuated vaccine induced effector CD4 T cells and opsonic antibodies against several O-antigens, the LPS core and to surface proteins of *P*. *aeruginosa*, providing protection against acute lethal pneumonia in mice [42]. These results reinforce the hypotheses that a successful vaccine for *P*. *aeruginosa* colonization prevention should induce multiple immune effectors.

There is evidence that an effective *P*. *aeruginosa* vaccine may require elicitation of both opsonizing antibodies, CD4^+^ T cells and IL-17 production to prevent infections. Thus, more recently, several preclinical studies leading to the induction of Th17-type cellular immunity are being pursued [41,44]. The antigens flagellin, pili, OMP or whole cell-based vaccines have been shown to be able to induce Th1 and Th17 immune responses. Recent studies also demonstrated that the inclusion of Th-17 promoting adjuvants was important for vaccine efficacy. Recently, these preclinical studies were thoroughly reviewed by [41,44].

As the use of antibiotics remains a challenge due to the high resistance to clinically available antibiotics, it is imperative to design new strategies to fight and eradicate Bcc infections [22]. In addition, the high variability of virulence within the complex and the fact that some virulence traits are strain-specific adds considerable difficulties in developing an effective vaccine [50]. However, no validated vaccines are yet available for any Bcc species. Nevertheless, several studies on the identification of Bcc putative immunogenic molecules and on the immune response resulting from Bcc infections have been carried out in the effort to develop a vaccine [50,51].

Some virulence factors such as proteases, exopolysaccharides and elastase have been studied as a target for a vaccine against Bcc infection (Figure 1), but so far none of them were considered protective (reviewed by Choh et al., 2013) [52]. Considerable research on vaccine development has been performed with the Bcc-related *Burkholderia pseudomallei* and *Burkholderia mallei*, including immunization with heat-killed bacteria, live-attenuated vaccines, subunit vaccines, glycoconjugates, DNA vaccines and viral vector-based vaccines, as reviewed by Wang. et al., 2020 [53]. However, the use of killed whole-cell vaccines against Bcc species has not been described until now.

Research to develop candidate vaccines to Bcc infections have been focused on live attenuated vaccines and subunit vaccines. To date, there is only one study regarding a live-attenuated vaccine for Bcc. Pradenas et al. (2017) showed that a live-attenuated vaccine using a *B*. *cenocepacia tonB* mutant conferred protection against an acute respiratory and lethal infection in a mouse model, with 87.5% survival rate by day 6 post-infection [54].

Subunit vaccines are composed of one or more purified microbial components, as antigenic proteins, epitopes or polysaccharides from the disease-causing agent, without their genetic material and with the addition of an adjuvant. A variety of antigens have been identified as promising candidates for subunit vaccines against *Burkholderia* species (Figure 1), including a few studies on Bcc bacteria [55,56,57]. The proteins that have been considered for this type of vaccine mainly include surface-exposed or outer membrane proteins, which are at the first line of contact with the immune system of the host and can be identified through available genomic information and bioinformatics tools [50]. For instance, a 17 kDa OmpA-like protein is known to confer protection to pulmonary colonization in mice. An intranasal vaccination with outer membrane proteins and the adjuvant adamantylamide dipeptide was first demonstrated to induce mucosal immunity against *B*. *multivorans,* to prevent the early stages of colonization and infection and to minimize tissue damage [58]. Later on, when administered with a mucosal nanoemulsion adjuvant, the 17 kDa OmpA was confirmed to induce a cross-neutralizing immunity to both *B*. *multivorans* and *B*. *cenocepacia*, accompanied by a balanced Th1/Th2 response, suggesting a potential cross-protection effect against other Bcc species [55]. Moreover, the OmpA-like protein BCAL2958 showed immunoreactivity with serum samples from CF patients infected with Bcc bacteria, resulting in the rise of IgG antibodies accompanied by an increase in the TNFα, elastase, hydrogen peroxide, nitric oxide and myeloperoxidase levels in neutrophils [56]. These studies suggest that the OmpA-like BCAL2958 is a good candidate for further studies both as an immunostimulant and as an immunoprotectant against Bcc infections. In addition, OmpW and linocin, two proteins required for attachment to host epithelial cells, were demonstrated to be protective antigens for mice infected with *B*. *cenocepacia* and *B*. *multivorans* [57]. Intraperitoneal immunization with these two antigens in BALB/c mice decreased the cellular load of these bacteria in the lung, which is explained by the Th1/Th2 responses. The ratio of antibodies in response to linocin indicated a bias towards a Th1 response, whereas for OmpW a mixed Th1/Th2 immune response was observed.

## 4. Immunotherapy Strategies against *P*. *aeruginosa* and *B*. *cepacia* Complex Infections

Passive immunotherapy has been revisited for treatment of antimicrobial resistant pathogens infections and for patients with an impaired immune system who cannot mount an effective immunity in response to active immunization [59,60]. For example, the use of monoclonal antibodies (MAbs) has been established in several therapeutic areas and represent an alternative or complement to antibiotic therapy, resulting in more rapid resolution of infections and shorter stays in intensive care units as well as reductions of morbidity, mortality and health care costs [61]. Generally, MAbs are derived from mice and genetically modified to improve tolerability in humans. Nevertheless, they still differ in glycosylation patterns from human antibodies, affecting their half-life and long-term tolerability. To overcome this problem, recently, a technique to utilize human B cells for the production of therapeutic MAbs was developed [62].

Several monoclonal antibodies and passive vaccines aimed at the neutralization of virulence factors of *P*. *aeruginosa* were developed (Table 2). One of the targets extensively used is the PcrV that is located at the tip of the type 3 secretion system (T3SS) injectisome complex of *P*. *aeruginosa,* being essential for the T3SS activity [63]. Clinical trials revealed that KB001-A, a PEGylated Fab fragment of an mouse anti-PcrV MAb had the ability to prevent *P*. *aeruginosa* ventilator-associated pneumonia [64]. However, low efficacy was observed in clinical trials using CF patients [63]. This observation can be in part due to the low levels of the T3SS proteins in sputum of CF patients chronically infected with *P*. *aeruginosa* [65]. Preclinical assays have shown that another anti-PcrV Mab, V2L2MD, had higher efficiency comparable to the KB001-A against *P*. *aeruginosa* infection in animal models [66].

Other targets studied were polysaccharides, such as the LPS (Table 2). LPS is a T-cell-independent antigen that triggers the innate immune system via the stimulation of pattern recognition receptors and the antibodies induced in response to them are mostly of the immunoglobulin M (IgM) isotype [67]. IgM antibodies have been used as therapeutic tools and have several favorable properties, such as their pentameric form providing 10 antigen binding sites, they bind antigens with high avidity, and are very effective complement activators [68]. This is the case of panobacumab, an IgM/κ isotype directed against the LPS O-polysaccharide moiety of *P*. *aeruginosa* serotype O11, that was successfully used in the intensive care units (ICU) as adjunctive therapy in patients with *Pseudomonas* VAP with a positive signal on clinical resolution [69].

MEDI3902, a bivalent bispecific mAb that selectively binds to both the PcrV protein and Psl exopolysaccharide on the surface of *P*. *aeruginosa* is being studied. Binding to PcrV on intact *P*. *aeruginosa* was shown to block T3SS injectisome-mediated cytotoxicity [70]. While, binding to Psl mediates opsonophagocytic killing of *P*. *aeruginosa* by host effector cells and inhibits the attachment to the host epithelial cells [70]. However, preliminary results of clinical trials on MV-ICU patients revealed that MED3902 were only effective in patients with lower baseline inflammation [71].

Preclinical studies, and clinical trials with egg yolk immunoglobulins (IgY)-based passive immunotherapies for the treatment of *P*. *aeruginosa* infections have been also performed [72]. IgY antibodies are serum immunoglobulin in birds, reptiles and amphibians that are transferred from serum to egg yolk to confer passive immunity to their embryos and offspring. The use of IgY based passive immunotherapies have the advantage of possible large-scale and cost-effective production; their extraction is performed directly from eggs, reducing animal harm and distress [73]. IgY have also reduced reactivity with mammalian factors, they do not activate the complement system, bind human Fc receptors that could mediate inflammatory response in the gastrointestinal tract or cross-react with human antibodies [74]. It was demonstrated that anti-*P*. *aeruginosa* IgY promotes formation of immobilized bacteria aggregates, enhancing bacterial killing by polymorphonuclear neutrophils (PMN)-mediated phagocytosis and thereby may facilitate a rapid bacterial clearance in the airways of CF patients [75]. One trial, NCT00633191, is based on the use by CF patients of daily mouthwash of IgY antibodies made in hens immunized with two different formaldehyde-fixed *P*. *aeruginosa* strains (PAO1 and Habs1). Although this study had a small sample size, after 12 years of daily treatment, a trend of a slightly later time of acquisition of a *P*. *aeruginosa* positive culture and later appearance of chronic infection was observed [72]. Another study was also performed, NCT01455675, a Phase III clinical trial of an IgY-based oral drug approved as an orphan drug against *P*. *aeruginosa* infections in CF patients. The results of its clinical efficacy are yet unclear, but no adverse immune or allergic reaction was observed (according to EudraCT report number 2011-000801-39) [76].

Bcc bacteria are known for their intrinsic resistance to a wide range of antimicrobials [80,81], being Bcc isolates from CF chronic infections, even more resistant when compared with non-CF isolates [18]. This discrepancy is likely linked to many factors present in CF patients, such as altered pharmacokinetics, failed drug delivery due to abnormally viscous bronchial secretions, high concentrations of organisms, biofilm formation and altered pH [82,83,84]. With this knowledge, alternatives to antibiotic therapies must be pursued, such as passive immunotherapies. An immunotherapy treatment uses the immune system of the patient to overcome or stop the bacterial infection, it can do this by boosting or changing the immune system allowing for a better immune response.

Unfortunately, no passive immunotherapy is currently available for Bcc bacteria. The majority of studies performed on passive immunization strategies were performed in the Bcc-related *B*. *pseudomallei* and *B*. *mallei* [51,85].

A work by Skurnik et al. used antibodies against the bacterial surface polysaccharide poly-β-(1-6)-*N*-acetyl-glucosamine (PNAG) and found that PNAG-specific antibodies confer protective immunity against all Bcc strains studied, that included *Burkholderia dolosa, B*. *multivorans* and *B*. *cenocepacia*. PNAG is a highly conserved surface polysaccharide produced both in vitro and in vivo by several bacteria, including Bcc bacteria. This surface polysaccharide mediates biofilm formation and is an important virulence factor. Skurnik et al. showed that antisera to PNAG mediated the opsonophagocytic killing of Bcc mediated by a human monoclonal antibody that has successfully undergone a phase 1 safety study [86]. The authors also showed in an intraperitoneal infection model in mice that the antibodies against PNAG could protect against lethal peritonitis from PNAG-producing Bcc bacteria, even during coinfection with methicillin-resistant *Staphylococcus aureus* [86].

Recently, Pimenta et al. (2021) described the use of an antibody-mediated passive protection (where an antibody against an antigen from the bacteria is used as a therapy), negatively affecting the infection process an allowing the host immune system to deal with the pathogen [87]. In this study a polyclonal goat antibody from a trimeric autotransporter adhesin (TAA) from *B*. *cenocepacia* K56-2 was shown to reduce the adhesion of the bacteria to bronchial epithelial cells, mucins, fibronectin, collagen type-I and also exhibited an inhibitory effect on the animal model of infection *Galleria mellonella*. Virulence assays performed for the first 72 h of infection showed that the antibody provided full protection against infection by *B*. *cenocepacia* K56-2. In contrast, weak protective effect was observed in *B*. *cenocepacia* J2315, while no protection was observed against *B*. *multivorans* VC13401, *Burkholderia contaminans* IST408 and *Escherichia coli* BL21-DE3 [87].

## 5. Novel Targets for Immunization and Immunotherapy Strategies against *P*. *aeruginosa* and Bcc Infections

The increasing extent of antibiotic resistances observed in an ever-growing number of bacteria is one of the most pressing matters related to bacterial infections and medicine in general. The development of new possible therapies for the treatment of antibiotic resistant bacteria is a crucial and enthusiastic growing field. Currently, no vaccines approved and commercially available for preemptive protection against *P*. *aeruginosa* or Bcc infections exist.

To unveil prospective novel vaccine candidates, exploitation of the genomic and proteomic information of *P*. *aeruginosa* through reverse-vaccinology has been performed by several research groups [88,89]. A schematic workflow of this approach is summarized in Figure 2.

Th17 cells have been described as a helper T cell lineage that can enhance antibacterial mucosal defenses and probably mediate protective vaccine-induced responses [28]. A reverse vaccinology approach was recently designed envisaging the discovery of Th17-stimulating protein antigens expressed by a plasmid library encoding *P*. *aeruginosa* proteins [89]. One of the proteins identified was the exoenzyme PopB that had strong T-cell epitopes. Immunization with purified PopB revealed enhanced clearance of *P*. *aeruginosa* from the lung and spleen after challenge in a murine model, protecting mice against lethal pneumonia in an antibody-independent fashion [89].

Recently, to gain more knowledge about the *P*. *aeruginosa* cell envelope, a combination of experimental proteomic data and reverse vaccinology, based on multidimensional protein identification technology (MudPIT) was performed [90]. This study revealed nine outer membrane proteins (OprE, OprI, OprF, OprD, PagL, OprG, PA1053, PAL and PA0833) as highly antigenic. Four of them (OprF, OprI, OprL and PA0833) are already under study as vaccine candidates [38,91,92].

In the case of Bcc bacteria, developments towards a protective vaccine are far behind when compared to *P*. *aeruginosa*, although some recent studies have reported various strategies to unveil possible antigens for the development of new therapeutic strategies.

An immunoproteomic approach was used by Shinoy et al., based on the comparison of the immunoproteomes of four strains of the two most clinically relevant species *B*. *cenocepacia* and *B*. *multivorans* [93]. The four strains were grown in Luria Bertani until the stationary phase was reached, and the total proteins were separated by a 2D-gel and examined by Western blot against a pool of seven sera from seven adult CF patients with a history of Bcc infection. The pooled sera were used to eliminate patient-specific effects. The proteins that reacted with the sera were then excised from the gel and identified by MALDI-TOF mass spectrometry. A total of 14 immunoreactive proteins exclusive to *B*. *cenocepacia,* 15 only present in *B*. *multivorans* and a total of 15 proteins across both species were identified [93]. Interestingly, only 5 out of the 14 proteins identified in both species are predicted to be located on the outer membrane, while those remaining were predicted to locate in the cytoplasm or the cytoplasmic membrane. Of the proteins identified in all four strains, RNAP had been previously shown to be immunogenic in other species and suggested to be associated with virulence, and also to contribute to the activation of invasion genes in *Salmonella enterica* [94,95,96]. This information, combined with the immunogenicity of RNAP in CF patients, suggests that the protein might play an active role in Bcc pathogenicity. Other proteins identified in all four strains included the elongation factor Tu, formerly identified as immunoreactive in the secretome of *B*. *cepacia* [97]. The identification of predicted cytoplasmatic immunoreactive proteins might indicate that these proteins are in contact with the host immune system due to bacterial secretion. In *B*. *pseudomallei*, this elongation factor Tu is secreted in outer membrane vesicles and has also been shown to reduce bacterial amounts when used to immunize mice [98]. The chaperonin GroEL was also present in the four strains tested. GroEL homologs from *B*. *pseudomallei* and *P*. *aeruginosa* had been identified as immunogenic [99,100] and have also been associated with intracellular invasion of *Legionella* [101]. This protein was able to protect against a lethal *Streptococcus pneumoniae* challenge and conferred protection against *Bacillus anthracis* mice infection [102,103].

A similar approach was performed by Sousa et al. to identify Bcc immunogenic proteins, based on 2D gel electrophoresis followed by mass spectrometry identification of the immunoreactive proteins against the pooled sera of three CF patients with a previous record of Bcc infection [104]. The authors used the *B*. *cenocepacia* J2315 strain, grown for 20 h in Petri plates containing artificial sputum medium at 37 °C under microaerophilic or aerobic conditions, to mimic environmental conditions of the lungs of CF patients [97]. A total of 24 proteins were identified as immunoreactive, with 19 of them being reported as immunogenic for the first time. Of the proteins identified, 10 were predicted to be extracytoplasmatic and found to be extremely conserved with the Bcc. Of the identified proteins two, BCAS0766 and BCAS0764, belong to the RND-2 multidrug efflux pump system [105]; a similar cluster has been studied in *B*. *pseudomallei* and demonstrated to be required for secretion of virulence factors and biofilm formation [106]. Another protein identified was the phage-shock protein PspA, this protein was found to be important for survival and virulence of different pathogens [107]. Namely, in *B*. *pseudomallei*, it was found to be essential for intracellular survival in a macrophage cell line [108].

Secreted proteins are known to play a role in initial phases of pathogenesis, being able to induce or elicit immune responses [109,110]. Therefore, another approach was performed for the identification of immunogenic secreted proteins in *B*. *cepacia* [97]. The strategy used was similar to those previously mentioned, where the culture supernatant was separated using 2D-gel electrophoresis, followed by Western blot analysis against three different sera, mice anti-outer membrane protein sera, mice anti-whole inactivated bacteria protein sera and mice anti-culture filtrate antigen sera. A set of 18 proteins were reactive with all three sera, and were suggested by Mariappan et al. (2009) as potential diagnostic markers or candidates for vaccine development against *B*. *cepacia* infections [97]. Of the 18 immunogenic proteins, 8 were involved in metabolism, 4 had roles in cellular processes, 5 in information storage and processing and 1 is a hypothetical protein. Only one of these proteins, a phospholipid/glycerol acyltransferase, was truly a secreted protein. These phospholipases were shown to play a role in bacterial survival and dissemination due to their potential to interfere with cellular signaling cascades and to modulate the host immune response [97].

Recently, Sousa et al. used a surface shaving approach to identify the surface exposed immunoproteome of *B*. *cenocepacia* J2315 [111]. Bioinformatic identification of putative surface-exposed proteins was combined with an experimental approach that consisted of the incubation of the live intact cells with trypsin, allowing the “shaving” of the surface proteins that were identified by liquid chromatography and mass spectrometry. Bacteria were cultivated in conditions that mimic those found in the CF lungs, namely, using artificial sputum growth medium (ASM) and microaerophilic atmosphere [112,113], 263 potential surface-exposed proteins were identified bioinformatically, with 143 of these having high probability of containing B-cell epitopes. The shaving approach was able to identify only 16 of the 143 potentially immunogenic proteins. The difference observed emphasize the importance of combined bioinformatics and experimental approaches [111]. The immunogenicity of three proteins identified in this work (BCAL2958, BCAL2645, BCAL2022) was demonstrated using sera samples from CF patients with previous history of Bcc infection, validating the approach for the detection of potential immunogenic proteins.

## 6. Conclusions

Either an effective *P*. *aeruginosa*, Bcc or combination of both, vaccine has been pursued for several decades. Many antigens have been studied and their based vaccines are under clinical trials or preclinical studies now, however there are currently none available for clinical application.

Recent developments in genomics, proteomics and bioinformatics have empowered researchers with novel tools to discover novel antigens that can be exploited for the development of vaccines. The number of complete and publicly available genomes from *P*. *aeruginosa* and Bcc is remarkable: as of 13 May 2021, 287 complete genomes and 2979 draft genomes are available at the Burkholderia Genome database [106], and 613 complete genomes and 9184 draft genomes are available at the Pseudomonas Genome BD [107]. These available data represent a huge opportunity for mining and discovery of novel antigens towards the development of novel immunotherapies to combat infections by these pathogens.

Passive immunotherapy trial results have shown to be advantageous for treating *P*. *aeruginosa* in patients unable to mount an effective immune response or after infection. For Bcc infections, passive immunotherapies are not explored. However, recently, several surface-exposed immunogenic antigens have been identified in Bcc bacteria that could be interesting targets for the development of passive immunotherapies.

Further knowledge of the molecular pathogenesis of *P*. *aeruginosa* and Bcc is required for the development of novel vaccines and therapies against these pathogens. The type of immunity that is essential for an effective Bcc or *P*. *aeruginosa* vaccine remains controversial, being the different immune responses depending on either the antigen tested, the host background or both. Therefore, the knowledge of the human immune response during infection with these bacterial pathogens is also important to design effective therapies. Thus, it is likely that effective vaccines against these pathogens will need to be tailored for specific patient populations, instead of a broad vaccine.

## Figures and Tables

**Figure 1 vaccines-09-00670-f001:**
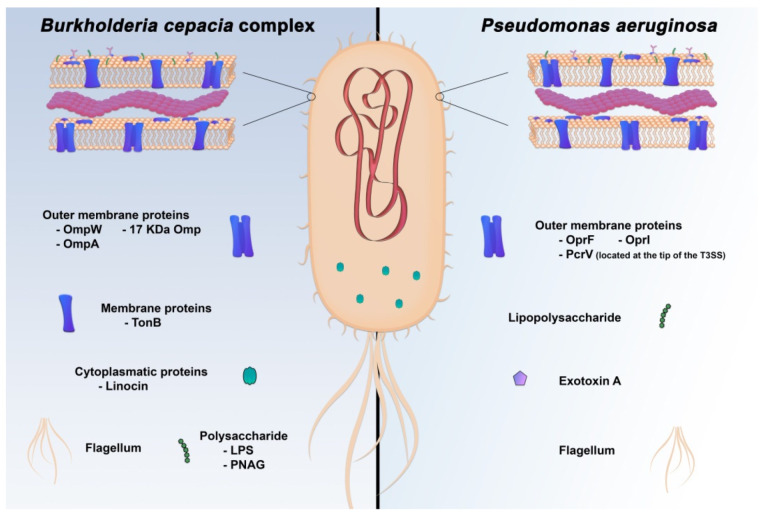
Antigens used in preclinical or clinical trials for development of either active, passive or in combination, vaccines against Bcc or *P*. *aeruginosa* infections.

**Figure 2 vaccines-09-00670-f002:**
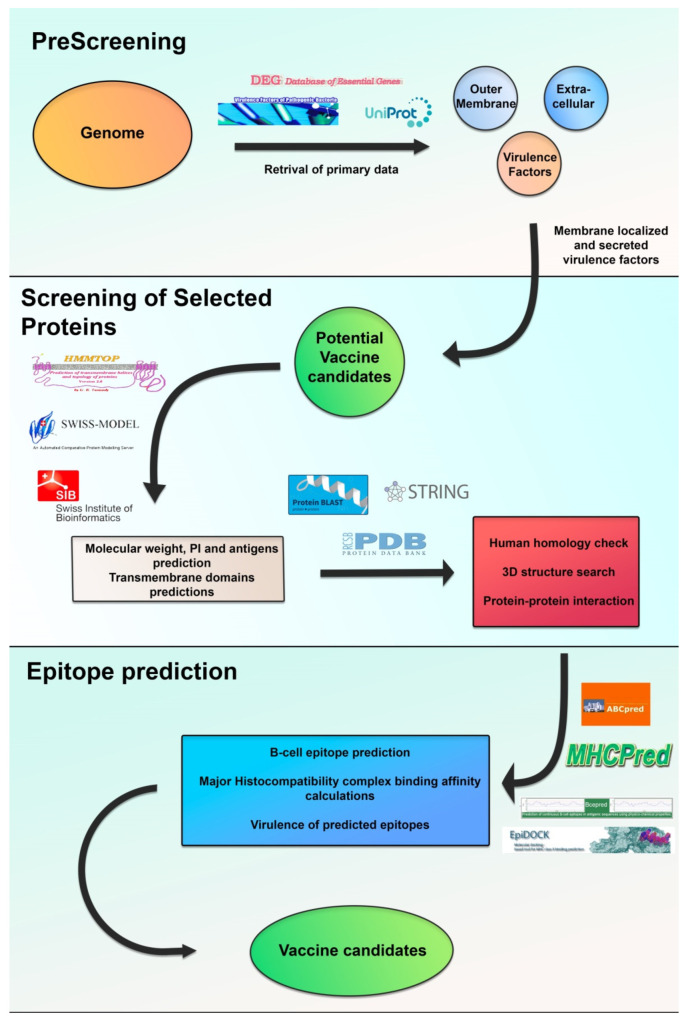
Reverse-vaccinology schematic workflow: Step (1)—Prescreening of primary data (e.g., genome) and prediction of subcellular localization, essentiality and virulence. Step (2)—Screening of the previous selected proteins for their immuno-protective potential, including the criteria molecular weight (MW) prediction, protein structural details and human homologue search. Step (3)—Epitope prediction using specific algorithms in order to obtain broad spectrum immunogenic peptides. The predicted epitopes are then screened for epitopes capable of binding higher numbers of MHC alleles in greater efficacy.

**Table 1 vaccines-09-00670-t001:** Active immunization clinical trials against *P*. *aeruginosa* infections.

Immunogen	Clinical Trial	Immunization Protocol	Results		References
			Pros	Cons	
Pseudostat (Paraformaldehyde-killed *P*. *aeruginosa*)	(Phase 1): Safety and immunogenicity test of a vaccine administered to healthy human subjects	Healthy volunteers (18–50 years of age) Oral dose (150 mg), 2 doses (day 0 and 28). Serological follow-up to 56-days post-vaccination.	Immunogenic in humans. Pooled sera collected after immunization had higher capacity to promote opsonophagocytotic killing (OPK) of *P*. *aeruginosa*.	Some neurological, gastrointestinal, and respiratory disorders were detected	[43]
Pseudogen (Heptavalent O-antigen)	(Phase 2): Efficacy evaluation of the vaccine in patients with acute leukemia and CF.	Intramuscular (IM) administration of 6–12 µg/kg.	Efficacy in preventing fatal *P*. *aeruginosa* infections in cancer and burn patients.	No benefit for leukemia and CF patients. Limited use due to their toxicity in 92% of patients tested.	[45]
PEV-01 (Polyvalent LPS extracts)	(Phase 2): Prospective, controlled study of a polyvalent vaccine in CF.	Three doses, SC, 1 month apart and the dose 4 after 1 year. 0.25 mL for CF patients under 12 years of age and 0.5 mL for patients over 12.		No benefit for CF patients.	[46]
Aerugen (Octavalent OPS-Toxin A conjugate)	(Phase 3): Analysis of the serological response after 10 years of repeated immunization of children with CF and efficacy on prevention of *P*. *aeruginosa* colonization.	Initial inoculations were given at 0, 2 and 12 months, and annual booster doses after the third year.	Increase of IgG levels to all vaccine components. Has a good safety profile for long-term use. The incidence of *P*. *aeruginosa* infection was lower compared with the non-vaccinated group.	Later unpublished results from a prospective trial of this vaccine in Europe did not show a delay in colonization and this vaccine was abandoned.	[47]
Bivalent FliC	(Phase 3): Immunization of CF patients not colonized with *P*. *aeruginosa* to evaluate its safety and efficacy.	Double-blind, placebo-controlled, multicenter trial. IM; 40 µg in CF patients (2–18 years of age), 4 doses, administrated each dose with 4 weeks apart and a booster dose after 1 year. Addition of adjuvant Al(OH)3, thiomersal.	Vaccine well tolerated. Active immunization of CF patients delayed the onset of chronic infection with *P*. *aeruginosa*, resulting in longer survival of these patients. High serum IgG titers to flagella vaccine subtypes.		[33]
CFC-101 (OMP extracted from 4 *P*. *aeruginosa* strains)	(Phase 2): Analysis of 2 immunization schedules of the OMP vaccine in burn patients.	Double-blind, randomized and placebo-controled trial. Adult patients with burn injuries in greater than 10% total body surface. IM; 3 doses (0.5 or 1.0 mg) with 3- or 7-day intervals.	The vaccine was safe and highly immunogenic in burn patients, especially with 1 mg doses at 3-day intervals.		[48]
IC43 (OprF/I)	NCT00778388 (Phase 1): Against *P*. *aeruginosa* in healthy volunteers	Placebo controlled, double-blind, multi-center, randomized trial. Four different doses (50–200 mcg) administered intramuscularly (IM) to healthy adults (18–65 years of age), with 2 doses given 7 days apart	No serious adverse effect. IC43 doses ≥50 mcg were sufficient to induce plateau of IgG antibody responses in healthy volunteers. At day 90, titers declined but remained higher than the placebo group for up to 6 months.	Higher doses, whether adjuvanted or not, were not more effective.	[37,38,49]
	NCT00876252 (Phase 2): Immunogenicity of IC43 in ICU admitted patients requiring MV	Patients were randomized to receive 3 different vaccine doses (100 mcg or 200 mcg IC43 with adjuvant, or 100 mcg without adjuvant) or placebo IM at days 0 and 7. Evaluation for 90 days.	At day 14 all IC43 administered groups had higher anti-OprF/I titers. Lower mortality in patients immunized with IC43 compared with placebo.	No statistical difference in *P*. *aeruginosa* infection rates between patients vaccinated with IC43 and placebo. However, most *P*. *aeruginosa* infections occurred before 14 days.	
	NCT01563263 (Phase 2/3): Confirmatory study assessing efficacy, immunogenicity and safety of IC43 vaccine in intensive care unit (ICU) patients	Placebo controlled, double-blind, multi-center, randomized trial. ICU patients requiring MV for more than 48 h, age 18–80 years. Patients were randomized to receive an IM injection of 100 mcg of IC43 or placebo on days 0 and 7.	Vaccine was well tolerated in the large population of medically ill, MV patients. The vaccine achieved high immunogenicity.	However, no clinical benefit over placebo was provided in terms of overall mortality.	

**Table 2 vaccines-09-00670-t002:** Passive immunization preclinical and clinical trials against *P*. *aeruginosa* infections.

Immunogen	Clinical Trial	Immunization Protocol	Results	References
Pros	Cons
KB001(anti-PcrV PEGylated mouse Mab)	Phase 1/2 (NCT00691587): Safety and pharmacokinetics (PK) of KB001 in mechanically ventilated (MV) ICU patients colonized with *P*. *aeruginosa*.	Patients (older than 18 years) will receive randomly either placebo, or single low-dose or single high-dose of KB001 intravenously (IV).	Safe and well-tolerated.	No anti-KB001 antibodies were detected.	[64,77]
Decrease in the incidence of *P*. *aeruginosa* associated pneumonia in patients on MV.
Phase 1/2 (NCT00638365): Dose escalation study of KB001 in CF patients colonized with *P*. *aeruginosa*.	Patients (older than 12 years) will randomly receive either placebo, single-dose 3 mg/kg or single-dose 10 mg/kg of KB001 IV.	Safe and well-tolerated.	No significant differences between KB001 and the placebo
Reduced lung inflammation of KB001 vaccinated patients.	group in *P*. *aeruginosa* colonization of CF patients.
KB001-A	Phase 2 (NCT01695343): Evaluation of the effect of KB001-A on time-to-need for antibiotic treatment of CF patients.	Patients (12–50 years of age) will randomly receive either placebo, KB001-A up to 5× IV at 10 mg/kg to a maximum dose of 800 mg per dose.	Safe and well-tolerated.	Reduced clinical efficacy, being not associated with an increased time to need for antibiotics.	[63]
(anti-PcrV PEGylated mouse MAb)	Modest FEV_1_ benefit and reduction in selected sputum inflammatory markers (IL-8).
(One amino acid difference from KB001)	
V2L2MD	Preclinical		Good prophylactic protection in several mouse models of *P*. *aeruginosa* infection.		[66]
(anti-PcrV Human MAb)
MEDI3902(anti-PcrV and Psl bispecific human MAb)	Phase 1 (NCT02255760): Safety evaluation, PK, anti-drug antibody (ADA) responses, ex vivo anticytotoxicity and OPK of MEDI3902 in healthy adults	Single IV infusion in healthy adults aged 18–60 years.	The safety and tolerability profile of MEDI3902 was acceptable.	Infusion-related reaction (e.g., skin rash).	[71,78]
Dose-escalation study: subjects were randomized in a 3:1 ratio to receive 250, 750, 1500 or 3000 mg of MEDI3902 or placebo.	Anti–*P. aeruginosa* activity was demonstrated in sera of treated subjects.
Subjects followed for 60 days afterwards.	
Phase 2 (NCT02696902): Evaluation of MEDI3902 efficacy and safety on the prevention of *P*. *aeruginosa* nosocomial pneumonia in MV patients	Participants will receive a single IV dose of placebo, MEDI3902 500 mg or MEDI3902 1500 mg.	Some clinical efficacy in ICU patients with lower baseline inflammation.	
Panobacumab or KBPA-101 or AR-101(IgM/κ isotype directed against the LPS O-polysaccharide moiety of *P*. *aeruginosa* serotype O11)	Phase 2: PK and safety profile of KBPA-101 in healthy volunteers		No adverse effects in healthy volunteers.		[61,69]
NCT00851435 (phase 2): Safety and PK in patients with hospital acquired pneumonia (HAP) caused by serotype O11 *P*. *aeruginosa*	HAP patients (older than 18 years of age) were treated by IV infusion of 1.2 mg/kg KBPA-101, 3 separate doses, every third day.	Improve clinical outcome in a shorter time.	
Passive immunotherapy targeting LPS can be a complementary strategy for the treatment of nosocomial *P*. *aeruginosa* pneumonia.
Aerucin or AR-105 (Human IgG1 MAb that targets *P*. *aeruginosa* alginate)	NCT02486770 (phase 1): Safety evaluation of Aerucin in healthy individuals.	IV administration up to 20 mg/kg monitored for 84 days in healthy individuals.	Safety up to doses of 20 mg/kg.		[79]
NCT03027609 (phase 2): Efficacy, safety and PK evaluation of Aerucin in combination with standard antibiotic treatment in *P*. *aeruginosa* VAP patients.	Placebo controlled, double-blind, randomized trial.		No significant difference between Aerucin and placebo patient groups for treatment of *P*. *aeruginosa* VAP patients.
Single IV infusion of Aerucin 20 mg/kg.
PseudIgY	NCT00633191 (phase 2): Study of anti-*pseudomonas* IgY in prevention of recurrence of *P*. *aeruginosa* infections in CF Patients.	Oral administration (gargle solution) of CF patients every night after toothbrushing.	After 12 years of prophylactic anti-*Pseudomonas* IgY treatment a reduction was observed in the level of infections with *P*. *aeruginosa* in the treated CF patients and no decrease in lung function.		[72]
(anti-*pseudomonas* IgY gargle)
PsAer-IgY(anti-*pseudomonas* IgY gargle)	NCT01455675 (phase 3): Evaluation of the clinical efficacy and safety of anti-*Pseudomonas* IgY in prevention of recurrence of *P*. *aeruginosa* infection in CF patients	Randomized, double-blind,	IgY antibodies were present in the oral cavity of treated patients for up to 24 h.	Clinical efficacy results were unclear.	[76]
placebo-controlled.	No adverse immune or allergic reaction.
Oral administration of CF patients (older than 5 years of age), every day with 70 mL gargling solution (contains 50 mg IgY) or placebo. Treatment for 24 months.

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
