# Peer review of "Immunization and Immunotherapy Approaches against Pseudomonas aeruginosa and Burkholderia cepacia Complex Infections"

_vaccines, 2021, doi:10.3390/vaccines9060670_

Round 1

Reviewer 1 Report

Overall this review is well written with some grammatical issues that can easily be fixed in the editing phase. The manuscript covers vaccine strategies against both Pseudomonas aeruginosa and strains in the Burkholderia cepacia complex. The review is very thorough, well organized and very descriptive. Only minor revisions are requested by this reviewer.

Organizational comment:

Figure 2 should be referred to earlier than the conclusions. Please move figure further up in the document.

Specific comments:

Line 45: diagnostic should be diagnosis 

Line 63: replace "a life threat" with "life threatening"

Line 66: replace "is" with "are"

Line 80: replace "being this" with "which is"

Line 120: no need to define lipopolysaccharide as  (LPS) as this was already done earlier

Line 124-126: Sentence doesn't make sense

Line 126: replace "is known" with "it is known"

Paragraph lines 123-128: add a sentence or so on how mucoid phenotype corresponds to virulence

Line 130: explain what the “a” or “b” subtypes are?

Line 138: "Outer membranes proteins" - should be "Outer membrane proteins" - membrane should not be plural

Line 139: replace "being considered" with "are being considered"

Line  147: replace "increase significantly" with "significantly increase"

Line 195: rewrite sentence containing "resulting in the evoke of IgG". What do the authors mean by 'evoke'??

Line 210: replace "have been" with "has been"

Line 273-275: rewrite sentence as there are grammar issues

Lines 290-295: sentence has grammar issues and is too long- break into two sentences.  Also what do authors mean when they say "accessed that antisera"?  

Line 354: replace "To notice is the fact that" with "Interestingly"

Line 360: replace "to play" with "may play"

Line 387: B. cepacia  should be in italics

Lines 390-391: expand this section. It is mentioned that 18 proteins were identified.  What were these proteins??  Are they similar or different to what the other studies identified?

Line 399 states conditions that mimicked the CF lung. What were these conditions?

Line 408: replace "have been" with "has been"

Lines 414/415: P. aeruginosa should be in italics

Line 418: represents should be represent

Author Response

Responses to Reviewer #1

Reviewer:

Overall this review is well written with some grammatical issues that can easily be fixed in the editing phase. The manuscript covers vaccine strategies against both Pseudomonas aeruginosa and strains in the Burkholderia cepacia complex. The review is very thorough, well organized and very descriptive. Only minor revisions are requested by this reviewer.

Answer: Thanks for the positive appreciation of our work. Just a remark to state that since our option was to include the two pathogens, it is quite impossible to make an exhaustive review on each system, and we opted to give an overview on both systems, retaining the most important advances in each one.

Reviewer: Organizational comment:

Figure 2 should be referred to earlier than the conclusions. Please move figure further up in the document.

Answer: We appreciate the suggestion. The figure was moved up in the document and now appears in section 3: Immunization strategies against P. aeruginosa and Bcc infections. As a consequence, it is now renumbered as Figure 1, and previous Figure 1 is now Figure 2.

Specific comments:

Reviewer: Line 45: “diagnostic” should be “diagnosis”

Answer: Done

Reviewer: Line 63: replace "a life threat" with "life threatening"

Answer: Done

Reviewer: Line 66: replace "is" with "are"

Answer: Done

Reviewer: Line 80: replace "being this" with "which is"

Answer: Done

Reviewer: Line 120: no need to define lipopolysaccharide as (LPS) as this was already done earlier

Answer: Done

Reviewer: Line 124-126: Sentence doesn't make sense

Answer: Thanks for the remark. The sentence was re-written and now reads as follows: “However, protection is only obtained for the specific O-antigen serotypes of the strains used [29].”

Reviewer: Line 126: replace "is known" with "it is known"

Answer: Done

Reviewer: Paragraph lines 123-128: add a sentence or so on how mucoid phenotype corresponds to virulence

Answer: Thanks for the suggestion. We have included the information requested:

See new lines:

“Mucoid strains produce the anionic exopolysaccharide alginate, which has been ascribed various functions related to the pathogenesis of P. aeruginosa in the CF lung. These include, among others, inhibition of phagocytosis by macrophages and neutrophils, suppression of neutrophil chemotaxis and opsonic antibody production, scavenging of hypochlorite and quenching of oxygen reactive species (reviewed in [31]). In addition, alginate contributes to biofilm formation, promoting the formation of microcolonies and contribute to antibiotic resistance [31].”

Reviewer: Line 130: explain what the “a” or “b” subtypes are?

Answer: Flagellin is the primary protein component of the flagellar filament, and it can be classified into two serotypes, a and b. We have slightly modified the sentence to incorporate the explanation. It now reads as follows: “In CF patients, the P. aeruginosa strains that colonize the patients are usually flagella-positive, which are classified as of the “a” or “b” serotypes  [32].”

Reviewer: Line 138: "Outer membranes proteins" - should be "Outer membrane proteins" - membrane should not be plural

Answer: Done

Reviewer: Line 139: replace "being considered" with "are being considered"

Answer: Done. We have added “and are being considered”

Reviewer: Line 147: replace "increase significantly" with "significantly increase"

Answer: Done

Reviewer: Line 195: rewrite sentence containing "resulting in the evoke of IgG". What do the authors mean by 'evoke'??

Answer: Thanks for the observation. We have substituted “evoke” by “rise”.

Reviewer: Line 210: replace "have been" with "has been"

Answer: Done

Reviewer: Line 273-275: rewrite sentence as there are grammar issues

Answer: The sentence was re-written and now reads as follows: ”The results of its clinical efficacy are yet unclear, but no adverse immune or allergic reaction was observed (according to the EudraCT report number 2011-000801-39)”.

Reviewer: Lines 290-295: sentence has grammar issues and is too long- break into two sentences. Also what do authors mean when they say "accessed that antisera"?  

Answer: Thanks for the suggestion. The sentence was re-written and divided in two. It now reads as follows: “Skurnik et al. showed that antisera to PNAG mediated the opsonophagocytic killing of Bcc mediated by a human monoclonal antibody that has successfully undergone a phase 1 safety study [86]. The authors also showed in an intraperitoneal infection model in mice that the antibodies against PNAG could protect against lethal peritonitis from PNAG-producing Bcc bacteria, even during coinfection with methicillin-resistant Staphylococcus aureus [86].”

Reviewer: Line 354: replace "To notice is the fact that" with "Interestingly"

Answer: Thanks for the suggestion. The sentence now reads as follows: “Interestingly, only 5 out of the 14 proteins identified in both species are predicted to be located on the outer membrane, while the remaining were predicted to locate in the cytoplasm or the cytoplasmic membrane.”

Reviewer: Line 360: replace "to play" with "may play"

Answer: Done

Reviewer: Line 387: B. cepacia should be in italics

Answer: Done

Reviewer: Lines 390-391: expand this section. It is mentioned that 18 proteins were identified.  What were these proteins??  Are they similar or different to what the other studies identified?

Answer: Thanks for the suggestion. We have included the information requested.

See new lines:

“Secreted proteins are known to play a role in initial phases of pathogenesis, being able to induce or elicit immune responses [110,111]. Therefore, another approach was performed for the identification of immunogenic secreted proteins in B. cepacia [98]. The strategy used was similar to those previously mentioned, where the culture supernatant was separated using 2D-gel electrophoresis, followed by Western blot analysis against 3 different sera, mice anti-outer membrane protein sera, mice anti-whole inactivated bacteria protein sera, and mice anti-culture filtrate antigen sera. A set of 18 proteins were reactive with all three sera, and were suggested by Mariappan et al. (2009) as potential diagnostic markers or candidates for vaccine development against B. cepacia infections [98]. Of the 18 immunogenic proteins, 8 were involved in metabolism, 4 had roles in cellular processes, 5 in information storage and processing and 1 is an hypothetical protein. Only one of these proteins, a phospholipid / glycerol acyltransferase, was truly a secreted protein. These phospholipases were shown to play a role in bacterial survival and dissemination due to their potential to interfere with cellular signaling cascades and to modulate the host immune response [98]. ”

Reviewer: Line 399 states conditions that mimicked the CF lung. What were these conditions?

Answer: The conditions were the use of Artificial Sputum Medium (ASM) and microaerophilic atmosphere. ASM medium is composed of, in g/L, porcine stomach mucin 5.0; low molecular-weight salmon sperm DNA 4.0; NaCl 5.0; KCl 2.2; casamino acids 5.0; Tris Base 1.81; and agar, 20 (pH 7.0), with 5.0 mL/L of egg yolk emulsion as a source of lecithin and 5.9 mg/L of the iron-chelating agent diethylene triamine pentaacetic acid (DTPA). The sentence was rewritten and now reads as follows: “Bacteria were cultivated in conditions that mimic those found in the CF lungs, namely using the Artificial Sputum growth Medium (ASM) and microaerophilic atmosphere [112, 113].”

Reviewer: Line 408: replace "have been" with "has been"

Answer: Done

Reviewer: Lines 414/415: P. aeruginosa should be in italics

Answer: Done

Reviewer: Line 418: represents should be represent

Answer: Done

Reviewer 2 Report

The authors summarized immunization and immunotherapy approaches against P. aeruginosa and B. cepacian complex infection. This study is particular interest, because the attempt matched a current era and is important solution for resistant pathogen. The authors should be considered for additional improvement.

Major

  1. The data regarding immunization and immunotherapy has been limited. However, several clinical trials and preclinical studies have been conducted. The authors should review an efficacy of each approach against two pathogens in a new section of “Effect of immunization and immunotherapy approach”. Moreover, the authors need to review side effects of each approach.
  2. The target which perform immunization or immunotherapy can be different, because there is a difference in mechanisms between each approach. The authors need to mention the targeted population.

Minor

  1. After performance of these approaches, how long does immune system against two pathogens continue?
  2. Please summarize a protocol of immunization of each study in Table 1 and 2.

Author Response

Responses to Reviewer #2

Reviewer #2

Reviewer:

The authors summarized immunization and immunotherapy approaches against P. aeruginosa and B. cepacia complex infection. This study is particular interest, because the attempt matched a current era and is important solution for resistant pathogen. The authors should be considered for additional improvement.

Major 1: The data regarding immunization and immunotherapy has been limited. However, several clinical trials and preclinical studies have been conducted.

Answer: We thank the comments. As suggested, additional information was added in Table 1 and 2.

We have also added the following information in section 3:

“There is evidence that an effective P. aeruginosa vaccine may require elicitation of both opsonizing antibodies, CD4+ T cells and IL-17 production to prevent infections. So, several pre-clinical studies leading to the induction of Th17-type cellular immunity are being pursued more recently [41,44]. The antigens flagellin, pili, OMP or whole cell-based vaccines have been shown to be able to induce Th1 and Th17 immune responses. Recent studies have also demonstrated that the inclusion of Th-17 promoting adjuvants was important for vaccine efficacy. Recently, these pre-clinical studies have been thoroughly reviewed by [41] and [44].”

Reviewer: Major 1: The authors should review an efficacy of each approach against two pathogens in a new section of “Effect of immunization and immunotherapy approach”.

Answer: We appreciate the suggestion. However, to the best of our knowledge, there is information on immunization strategies against both pathogens, and therefore we cannot add the suggested new section.  

Reviewer: Major 1: Moreover, the authors need to review side effects of each approach.

Answer: Thanks for the suggestion. I order to accommodate this suggestion, Tables 1 and 2 were revised and now include Pros and Cons for each clinical trial. Please see revised Tables 1 and 2.

Reviewer: Major 2: The target which performs immunization or immunotherapy can be different, because there is a difference in mechanisms between each approach. The authors need to mention the targeted population.

Answer: Thanks for the suggestion, which we think refers to the different types of target cells. This information when available, was added to the revised Tables 1 and 2.

Reviewer: Minor 1: After performance of these approaches, how long does immune system against two pathogens continue?

Answer: We appreciate the question, but we were unable to obtain information on this important issue.

Reviewer: Minor 2: Please summarize a protocol of immunization of each study in Table 1 and 2.

Answer: Thanks for the suggestion. The protocol of immunization of each study was introduced in the revised Tables 1 and 2.

Round 2

Reviewer 2 Report

The authors revised appropriately. No further correction is necessary.